# Flow Synthesizer: Universal Audio Synthesizer Control with Normalizing Flows †

**Philippe Esling** [1,*] , **Naotake Masuda** [1], **Adrien Bardet** [1], **Romeo Despres** [1] **and Axel Chemla-Romeu-Santos** [1,2]

1    IRCAM—CNRS UMR 9912 Sorbonne Université, 75004 Paris, France; napsau23@gmail.com (N.M.); adrien.bardet@live.fr (A.B.); despres.romeo@gmail.com (R.D.); chemla@ircam.fr (A.C.-R.-S.)
2    Laboratorio d'Informatica Musicale (LIM), UNIMI, 20133 Milano, Italy
*    Correspondence: esling@ircam.fr
†    This paper is an extended version of the conference paper presented at the 22nd International Conference on Digital Audio Effects (DaFX), Birmingham, UK, 2–6 September 2019.

**Abstract:** The ubiquity of sound synthesizers has reshaped modern music production, and novel music genres are now sometimes even entirely defined by their use. However, the increasing complexity and number of parameters in modern synthesizers make them extremely hard to master. Hence, the development of methods allowing to easily create and explore with synthesizers is a crucial need. Recently, we introduced a novel formulation of audio synthesizer control based on learning an organized latent audio space of the synthesizer's capabilities, while constructing an invertible mapping to the space of its parameters. We showed that this formulation allows to simultaneously address *automatic parameters inference*, *macro-control learning*, and *audio-based preset exploration* within a single model. We showed that this formulation can be efficiently addressed by relying on Variational Auto-Encoders (VAE) and Normalizing Flows (NF). In this paper, we extend our results by evaluating our proposal on larger sets of parameters and show its superiority in both parameter inference and audio reconstruction against various baseline models. Furthermore, we introduce *disentangling flows*, which allow to learn the invertible mapping between two separate latent spaces, while steering the organization of some latent dimensions to match target variation factors by splitting the objective as partial density evaluation. We show that the model disentangles the major factors of audio variations as latent dimensions, which can be directly used as *macro-parameters*. We also show that our model is able to learn semantic controls of a synthesizer, while smoothly mapping to its parameters. Finally, we introduce an open-source implementation of our models inside a real-time Max4Live device that is readily available to evaluate creative applications of our proposal.

**Keywords:** audio synthesizer; normalizing flows; variational inference; music information retrieval; machine learning; probabilistic graphical models; generative models; creative AI

## 1. Introduction

Synthesizers are parametric systems able to generate audio signals ranging from musical instruments to entirely unheard-of sound textures. Since their commercial beginnings more than 50 years ago, synthesizers have revolutionized music production, while becoming increasingly accessible, even to neophytes with no background in signal processing.

While there exists a variety of sound synthesis types [1], all of these techniques require an extensive a priori knowledge to make the most out of a synthesizer possibilities. Hence, the main appeal of these systems (namely their versatility provided by large sets of parameters) also entails their major drawback. Indeed, the sheer combinatorics of parameter settings makes exploring all possibilities

to find an adequate sound a daunting and time-consuming task. Furthermore, there exist highly non-linear relationships between the parameters and the resulting audio. Unfortunately, no synthesizer provides intuitive controls related to perceptual and semantic properties of the generated audio. Hence, a method allowing an intuitive and creative exploration of sound synthesizers has become a crucial need, especially for non-expert users.

A potential direction taken by synth manufacturers is to propose programmable *macro-controls* that allow to efficiently manipulate the generated sound qualities by controlling multiple parameters through a single knob. However, these need to be programmed manually, which still requires expert knowledge. Furthermore, no method has ever tried to tackle this *macro-control learning* task, as this objective appears unclear and depends on a variety of unknown factors. An alternative to manual parameters setting would be to infer the set of parameters that could best reproduce a given *target sound*. This task of *parameters inference* has been studied in the past years using various techniques, such as iterative relevance feedback on audio descriptors [2], Genetic Programming to directly grow modular synthesizers [3], or bi-directional LSTM with highway layers [4] to produce parameters approximation. Although these approaches might be appealing, they all share the same fundamental flaws that (i) though it is unlikely that a synthesizer can generate exactly any audio target, none explicitly model these limitations, (ii) they do not account for the non-linear relationships that exist between parameters and the corresponding synthesized audio, and (iii) none of these approaches allow for higher-level controls or interaction with audio synthesizers. Hence, no approach has succeeded in unveiling the true relationships between these *auditory* and *parameters* spaces. Hence, it appears mandatory to organize the parameters and audio capabilities of a given synthesizer in their respective spaces, while constructing an invertible mapping between these spaces in order to access a range of high-level interactions. This idea is depicted in Figure 1.

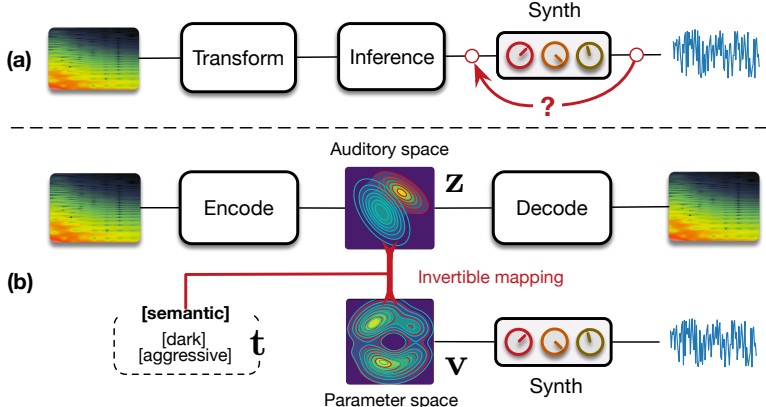

**Figure 1.** *Universal synthesizer control.* (**a**) Previous methods perform direct parameter inference from the audio, which is inherently limited by the non-differentiable synthesis operation and provides no higher-level form of control. (**b**) Our novel formulation states that we should first learn an organized and compressed latent space **z** of the synthesizer's audio capabilities, while mapping it to the space **v** of its synthesis parameters. This provides a deeper understanding of the principal dimensions of audio variations in the synthesizer, and an access to higher-level interactions.

The recent rise of *generative models* might provide an elegant solution to these questions. Indeed, amongst these models, the *Variational Auto-Encoder* (VAE) [5] aims to uncover the underlying structure of the data, by explicitly learning a *latent space* [5]. This space can be seen as a high-level representation, which aims to disentangle underlying variation factors and reveal interesting structural properties of the data [5,6]. VAEs address the limitations of control and analysis through this latent space, while being able to learn on small sets of examples. Furthermore, the recently proposed *Normalizing*

*Flows* (NF) [7] also allow to model highly complex distributions in this latent space. Although the use of VAEs for audio applications has only been scarcely investigated, Esling et al. [8] recently proposed a perceptually regularized VAE that learns a space of audio signals aligned with perceptual ratings via a regularization loss. The resulting space exhibits an organization that is well aligned with perception. Hence, this model appears as a valid candidate to learn an organized audio space.

Recently, we introduced a radically novel formulation of audio synthesizer control [9] by formalizing it as the general question of finding an invertible mapping between organized latent spaces, linking the audio space of a synthesizer's capabilities to the space of its parameters. We provided a generic probabilistic formalization and showed that it allows to address simultaneously the tasks of *parameter inference*, *macro-control learning*, and *audio-based preset exploration* within a single model. To solve this new formulation, we proposed *conditional regression flows*, which map a latent space to any given target space, as depicted in Figure 2. Based on this formulation, *parameter inference* simply consisted of encoding the audio target to the latent audio space that is mapped to the parameter space. Interestingly, this bypasses the well-known blurriness issue in VAEs as we can generate directly with the synthesizer instead of the decoder. In this paper, we extend the evaluation of our proposal on larger sets of parameters against various baseline models and show its superiority in parameter inference and audio reconstruction. Furthermore, we discuss how our model is able to address the task of automatic *macro-control learning* that we introduced in Ref. [9] with this increased complexity. As the latent dimensions are continuous and map to the parameter space, they provide a natural way to learn the perceptually most significant macro-parameters. We show that these controls map to smooth, yet non-linear parameters evolution, while remaining perceptually continuous. Hence, this provides a way to learn the compressed and principal dimensions of macro-control in a synthesizer. Furthermore, as our mapping is invertible, we can map synthesis parameters back to the audio space. This allows intuitive *audio-based preset exploration*, where exploring the neighborhood of a preset encoded in the audio space yields similarly sounding patches, yet with largely different parameters. In this paper, we further propose *disentangling flows* to steer the organization of some of the latent dimensions to match given target distributions. We evaluate the ability of our model to learn these *semantic controls* by explicitly targeting disentanglement in the latent space of the semantic tags associated to synthesizer presets. We show that, although the model learns to separate the semantic distributions, the corresponding controls are not easily interpretable. Finally, we introduce a real-time implementation of our model in *Ableton Live* and discuss its potential use in creative applications (All code, supplementary figures, results, and the real-time Max4Live plugin are available as open-source packages on a supporting webpage: https://acids-ircam.github.io/flow_synthesizer/).

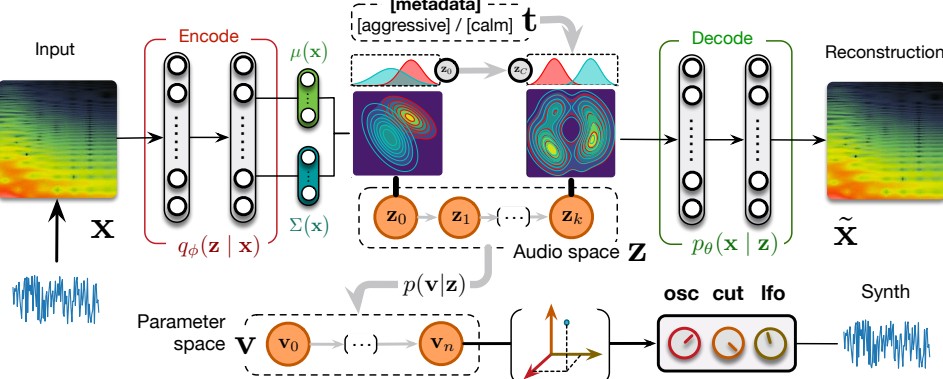

**Figure 2.** *Universal synthesizer control*. We learn an organized latent audio space **z** of a synthesizer capabilities with a Variational Auto-Encoder (VAE) parameterized with Normalizing Flow (NF). This space maps to the parameter space **v** through our proposed *regression flow* and can be further organized with metadata targets **t**. This provides sampling and invertible mapping between different spaces.

## 2. State-Of-Art

### 2.1. Generative Models and Variational Auto-Encoders

*Generative models* aim to understand a given set of input examples $\mathbf{x} \in \mathbb{R}^{d_x}$ by modeling the underlying probability distribution of the data $p(\mathbf{x})$. To do so, we introduce *latent variables* defined in a lower-dimensional space $\mathbf{z} \in \mathbb{R}^{d_z}$ ($d_z \ll d_x$). These variables can be seen as a higher-level representation that could have led to generate a given example. The complete model is then defined by the joint distribution $p(\mathbf{x}, \mathbf{z}) = p(\mathbf{x}|\mathbf{z})p(\mathbf{z})$. In order to obtain $p(\mathbf{x})$, we would need to marginalize $\mathbf{z}$ from the joint probability as follows

$$p(\mathbf{x}) = \int p(\mathbf{x} \mid \mathbf{z})p(\mathbf{z})d\mathbf{z}. \tag{1}$$

Unfortunately, as real-world data follow complex distributions, this formulation usually cannot be solved analytically. The idea of *variational inference* (VI) is to solve this problem through *optimization* by assuming a simpler approximate distribution $q_\phi(\mathbf{z}|\mathbf{x}) \in \mathcal{Q}$ from a family of parametric densities [10], where $\phi$ denotes the variational parameters that we can optimize. The goal of VI is to minimize the difference between this approximation and the real distribution, by minimizing the Kullback–Leibler (KL) divergence between these densities

$$q_\phi^*(\mathbf{z}|\mathbf{x}) = \text{argmin}_{q_\phi(\mathbf{z}|\mathbf{x}) \in \mathcal{Q}} \mathcal{D}_{KL}\big[q_\phi(\mathbf{z}|\mathbf{x}) \parallel p(\mathbf{z}|\mathbf{x})\big]. \tag{2}$$

However, our original problem is that we did not have a closed-form solution to the posterior $p(\mathbf{z} \mid \mathbf{x})$. By developing this KL divergence, re-arranging terms (the detailed development can be found in [5]) and introducing parametric distributions for the likelihood $p_\theta(\mathbf{x} \mid \mathbf{z})$ and prior $p_\theta(\mathbf{z})$, we obtain

$$\log p(\mathbf{x}) - \mathcal{D}_{KL}\big[q_\phi(\mathbf{z}|\mathbf{x}) \parallel p(\mathbf{z}|\mathbf{x})\big] = \mathbb{E}_{\mathbf{z}}\big[\log p(\mathbf{x}|\mathbf{z})\big] - \mathcal{D}_{KL}\big[q_\phi(\mathbf{z}|\mathbf{x}) \parallel p(\mathbf{z})\big]. \tag{3}$$

This formulation describes the quantity we want to model $\log p(\mathbf{x})$ minus the error we make by using an approximate $q$ instead of the true $p$. Therefore, we can optimize this alternative objective, called the *evidence lower bound* (ELBO), by optimizing the parameters $\phi$ and $\theta$ of the distributions

$$\mathcal{L}_{\theta,\phi} = \mathbb{E}\big[\log p_\theta(\mathbf{x}|\mathbf{z})\big] - \beta \cdot \mathcal{D}_{KL}\big[q_\phi(\mathbf{z}|\mathbf{x}) \parallel p_\theta(\mathbf{z})\big]. \tag{4}$$

Intuitively, the ELBO minimizes the reconstruction error through the likelihood of the data given a latent $\log p_\theta(\mathbf{x}|\mathbf{z})$, while regularizing the distribution $q_\phi(\mathbf{z}|\mathbf{x})$ to follow a given prior distribution $p_\theta(\mathbf{z})$. We can see that this equation involves $q_\phi(\mathbf{z}|\mathbf{x})$ which *encodes* the data $\mathbf{x}$ into the latent representation $\mathbf{z}$ and a *decoder* $p_\theta(\mathbf{x}|\mathbf{z})$, which generates $\mathbf{x}$ given a $\mathbf{z}$. This structure defines the *Variational Auto-Encoder* (VAE), where we can use parametric neural networks to model the *encoding* ($q_\phi$) and *decoding* ($p_\theta$) distributions. VAEs are powerful representation learning frameworks, while remaining simple and fast to learn without requiring large sets of examples [11].

However, the original formulation of the VAE entails several limitations. First, it has been shown that the KL divergence regularization can lead both to uninformative latent codes (also called *posterior collapse*) and variance over-estimation [12]. One way to alleviate this problem is to rely on the *Maximum Mean Discrepancy* (MMD) instead of the KL to regularize the latent space, leading to the WassersteinAE (WAE) model [13]. Second, one of the key aspect in the success of VI lies in the choice of the family of approximations. The simplest choice is the *mean-field* family where latent variables are mutually independent and parametrized by distinct variational parameters $q(\mathbf{z}) = \prod_{j=1}^m q_j(z_j)$. Although this provide an easy tool for analytical development, it might prove too simplistic when modeling complex data as this assumes pairwise independence among every latent axis. In order to alleviate this issue, normalizing flows [7] have been proposed by adding a sequence of invertible transformations to the latent variable, providing a more expressive inference process.

### 2.2. Normalizing Flows

In order to transform a probability distribution, we can rely on the *change of variable* theorem. As we deal with probability distributions, we need to *scale* the transformed density so that it still sums to one, which is measured by the Jacobian of the transform. Formally, let $\mathbf{z} \in \mathbb{R}^d$ be a random variable with distribution $q(\mathbf{z})$ and $f : \mathbb{R}^d \rightarrow \mathbb{R}^d$ an invertible smooth mapping. We can use $f$ to transform $\mathbf{z} \sim q(\mathbf{z})$, so that the resulting random variable $\mathbf{z}' = f(\mathbf{z})$ has the following probability distribution

$$q(\mathbf{z}') = q(\mathbf{z}) \left| \det \frac{\partial f^{-1}}{\partial \mathbf{z}'} \right| = q(\mathbf{z}) \left| \det \frac{\partial f}{\partial \mathbf{z}} \right|^{-1} \tag{5}$$

where the last equality is obtained through the inverse function theorem [7]. As we can see, this allows us to perform inference by relying on a more complicated (transformed) distribution $q(\mathbf{z}')$, while still being able to keep the simplicity of a mathematical development based on $q(\mathbf{z})$. Now, we can iteratively apply this reasoning and perform an arbitrary number of transforms to our original variable such that $\mathbf{z}_k = f_1 \circ ... \circ f_k(\mathbf{z}_0)$, in order to obtain a final distribution $\mathbf{z}_k \sim q_k(\mathbf{z}_k)$ given by

$$\begin{aligned} q_k(\mathbf{z}_k) &= q_0(f_1^{-1} \circ ... \circ f_k^{-1}(\mathbf{z}_k)) \prod_{i=1}^{k} \left| \det \frac{\partial f_i^{-1}}{\partial \mathbf{z}_i} \right| \\ &= q_0(\mathbf{z_0}) \prod_{i=1}^{k} \left| \det \frac{\partial f_i}{\partial \mathbf{z}_{i-1}} \right|^{-1} \end{aligned} \tag{6}$$

This series of transformations, called a *normalizing flow* [7], can turn a simple distribution into a complicated multimodal density. For practical use of these flows in inference, we need to define transforms whose Jacobian determinants are easy to compute. Interestingly, *Auto-Regressive* (AR) transforms fit this requirement as they lead to a triangular Jacobian matrix. Different types of AR flows were proposed such as *Inverse AR Flows* (IAF) [14] and *Masked AR Flows* (MAF) [15]. These flows allow to introduce dependencies between different dimensions of the original random variables.

Normalizing Flows in VAEs

Normalizing flows allow to address the simplicity of variational approximations by complexifying their posterior distribution [7]. In the case of VAEs, we parameterize the approximate posterior distribution with a flow of length $K$, $q_\phi(\mathbf{z}|\mathbf{x}) = q_K(\mathbf{z}_K)$, and the new optimization loss can be simply written as an expectation over the initial distribution $q_0(\mathbf{z})$

$$\begin{aligned} \mathcal{L} &= \mathbb{E}_{q_\phi(\mathbf{z}|\mathbf{x})} \left[ \log q_\phi(\mathbf{z}|\mathbf{x}) - \log p(\mathbf{x}, \mathbf{z}) \right] \\ &= \mathbb{E}_{q_0(\mathbf{z}_0)} \left[ \ln q_0(\mathbf{z}_0) \right] - \mathbb{E}_{q_0(\mathbf{z}_0)} \left[ \log p(\mathbf{x}, \mathbf{z}_K) \right] - \mathbb{E}_{q_0(\mathbf{z}_0)} \left[ \sum_{i=1}^{k} \log \left| \det \frac{\partial f_i}{\partial \mathbf{z}_{i-1}} \right| \right] \end{aligned} \tag{7}$$

The resulting objective can be easily optimized since $q_0(\mathbf{z})$ is still a Gaussian distribution from which we can easily sample. However, the final samples $\mathbf{z}_k$ used by the decoder are drawn from the much more complex transformed distribution.

### 2.3. Synthesizer Parameters Optimization

In the past years, the automatic parameterization of synthesizers has been the subject of several studies [3,4,16]. All of these approaches share the objective to optimize the correspondence between the generated sound and a given target sound. In the approach proposed by Cartwright et al. [2], audio descriptors such as the *Mel Frequency Cepstral Coefficients* (MFCCs) are used to evaluate the perceptual similarity to the target sound. This similarity is then iteratively refined during the search phase by weighting the different descriptors based on relevance feedback provided by the user [2]. Although this approach allow for user interaction, it seems to be very inaccurate and slow. Several

approaches were proposed based on genetic algorithms and Genetic Programming (GP) [17] in order to automatically construct modular synthesizer patches that approximate a given target sound. The heuristic is conditioned on a set of input control functions (target amplitude and frequency over time). The approach proved quite successful, managing to retrieve complicated frequency modulation sounds with high precision. Yet, the main limitation of this system is that it induce very high computation times, with as much as 10 to 200 h required to produce a single audio approximation. Hence, this renders the approach unusable in realistic studio or stage contexts. Roth et al. [16] compared different optimization techniques, namely *genetic programming*, iterative *hill-climbing*, a single-layer artificial *neural network* and a simple nearest neighbour algorithm performed on a grid sampling of the parameter space of the synthesizer. In this study, the genetic programming approach appears to provide the best results. Very recently, Yee-King et al. [4] tackled the same problem by using more advanced recurrent neural networks (bidirectional LSTMs). However, the number of parameters and complexity of the sounds studied remains in a quite low setting.

All of these approaches share the same flaws that they do not account for the non-linear relationships that exist between parameters and the corresponding synthesized audio, nor do they provide higher-level controls than the target-based parameters inference task. Here, we argue that it is mandatory to unveil the relationships between the *auditory* and *parameters* spaces of a synthesizer, and show that it provides multiple forms of high-level control.

## 3. Our Proposal

### 3.1. Formalizing Synthesizer Control

Considering a dataset of audio samples $\mathcal{D} = \{\mathbf{x}_i\}, i \in [1, n]$ where the $\mathbf{x}_i \in \mathbb{R}^d$ follow an unknown distribution $p(\mathbf{x})$, we can introduce latent factors $\mathbf{z} \in \mathbb{R}^z$ to model the joint distribution $p(\mathbf{x}, \mathbf{z}) = p(\mathbf{x} \mid \mathbf{z}) p(\mathbf{x})$ as detailed in Section 2.1. In our case, some $\bar{\mathbf{x}} \in \mathcal{D}_s \subset \mathcal{D}$ inside this set have been generated by a given synthesizer. This synthesizer defines a generative function $f_s(\mathbf{v}; p, i) = \bar{\mathbf{x}}$ where $\mathbf{v} \in \mathbb{R}^s$ is a set of parameters that produce $\bar{\mathbf{x}}$ at a given pitch $p$ and intensity $i$. However, in the general case, we know that if $\mathbf{x}_j \notin \mathcal{D}_s$, then $\mathbf{x}_j = f_s(\mathbf{v}) + \epsilon$ where $\epsilon$ models the error made when trying to reproduce an arbitrary audio sample $\mathbf{x}_j$ with a given synthesizer. Finally, we consider that some audio examples are annotated with a set of *categorical semantic tags* $\mathbf{t}_i = \{0, 1\}$, which define high-level perceptual properties that separate *unknown* latent factors $\mathbf{z}$ and *target* factors $\mathbf{t}$. Hence, the complete generative story of a synthesizer can be defined as

$$p(\mathbf{x}, \mathbf{v}, \mathbf{t}, \mathbf{z}) = p(\mathbf{x}|\mathbf{v}, \mathbf{t}, \mathbf{z}) p(\mathbf{v}|\mathbf{t}, \mathbf{z}) p(\mathbf{t}|\mathbf{z}) p(\mathbf{z}). \tag{8}$$

This very general formulation entails our original idea that we should uncover the relationship between the latent audio $\mathbf{z}$ and parameters $\mathbf{v}$ spaces by modeling $p(\mathbf{v}, \mathbf{z})$. The advantage of this formulation is that the reduced dimensionality $\mathbb{R}^z \ll \mathbb{R}^x$ of the latent $\mathbf{z}$ simplifies the problem of parameters inference, by relying on a more adequate and smaller input space. Furthermore, this formulation also provides a natural way of learning *macro-controls* by inferring $p(\mathbf{v}|\mathbf{z})$, where separate dimensions of $\mathbf{z}$ are expected to produce smooth auditory transforms. Interestingly, this can be seen as a way to learn the principal dimensions of audio variations in the synthesizer

### 3.2. Mapping Latent Spaces with Regression Flows

In order to map the latent $\mathbf{z}$ and parameter $\mathbf{v}$ spaces, we can first consider that the latent $\mathbf{z}$ and semantic $\mathbf{t}$ variables are both unknown latent factors where $p(\mathbf{z}') = p(\mathbf{z}, \mathbf{t})$. Hence, we can first address the following reduced formulation

$$\log p_\theta(\mathbf{x}, \mathbf{v}, \mathbf{z}') = \log(p_\theta(\mathbf{x}|\mathbf{v}, \mathbf{z}') p_\theta(\mathbf{z}')) + \log p_\theta(\mathbf{v}|\mathbf{z}'). \tag{9}$$

This allows to separately model the variational approximation (detailed in Section 2.1), while solving the inference problem $p_\theta(\mathbf{v}|\mathbf{z})$. To address this inference, we propose to optimize the parameters $\boldsymbol{\psi}$ of a transform $f_{\boldsymbol{\psi}}$ so that $\mathbf{v} = f_{\boldsymbol{\psi}}(\mathbf{z}) + \boldsymbol{\epsilon}$, where $\boldsymbol{\epsilon} \sim \mathcal{N}(\mathbf{0}, \mathbf{C}_v)$ models the inference error as a zero-mean additive Gaussian noise with covariance $\mathbf{C}_v$. Here, we assume that the covariance decomposes into $\mathbf{C}_v^{-1} = \sum_i \exp(\lambda_i)\mathbf{Q}_i$, where $\mathbf{Q}_i$ are fixed basis functions and the $\lambda$ are hyperparameters. Therefore, the full joint likelihood that we will optimize is given by

$$\mathcal{L}_{f_{\boldsymbol{\psi}}, \lambda} = \log \left[ p_\theta(\mathbf{v}|f_{\boldsymbol{\psi}}, \lambda, \mathbf{z}) p_\theta(f_{\boldsymbol{\psi}}|\mathbf{z}) p_\theta(\lambda|\mathbf{z}) \right]. \tag{10}$$

If we know the optimal transform $f_{\boldsymbol{\psi}}$ and parameters $\lambda$, the likelihood of the data is

$$p_\theta(\mathbf{v} \mid f_{\boldsymbol{\psi}}, \lambda, \mathbf{z}) = \mathcal{N}(\mathbf{v}; f_{\boldsymbol{\psi}}(\mathbf{z}), \mathbf{C}_v) \tag{11}$$

However, the two posteriors $p_\theta(f_{\boldsymbol{\psi}}|\mathbf{z})$ and $p_\theta(\lambda|\mathbf{z})$ remain intractable in the general case. In order to solve this issue, we rely again on variational inference by defining an approximation $q_\phi(f_{\boldsymbol{\psi}}, \lambda|\mathbf{v}, \mathbf{z})$ and assume that it factorizes as $q(f_{\boldsymbol{\psi}}, \lambda|\mathbf{v}, \mathbf{z}) = q(f_{\boldsymbol{\psi}}|\mathbf{v}, \mathbf{z}) q(\lambda|\mathbf{v}, \mathbf{z})$. Therefore, our complete inference is

$$\mathcal{L}_{f_{\boldsymbol{\psi}}, \lambda} = \log \left[ p_\theta(\mathbf{v}|f_{\boldsymbol{\psi}}, \lambda, \mathbf{z}) \right] + \mathcal{D}_{KL} \left[ q_\phi(f_{\boldsymbol{\psi}}|\mathbf{z}, \mathbf{v}) \| p_\theta(f_{\boldsymbol{\psi}}|\mathbf{z}) \right] + \mathcal{D}_{KL} \left[ q_\phi(\lambda|\mathbf{z}, \mathbf{v}) \| p_\theta(\lambda|\mathbf{z}) \right] \tag{12}$$

Hence, we can optimize our approximations through the KL divergence if we find a closed form. To solve for $\lambda$, we use a Gaussian distribution for both the prior $p_\theta(\lambda|\mathbf{z}) = \mathcal{N}(\lambda, \mu_\lambda, C_\lambda)$ and posterior $q_\phi(\lambda|\mathbf{z}, \mathbf{v}) = \mathcal{N}(\lambda, \mu_q, C_q)$. Hence, we obtain a simple analytical solution for $\lambda$. However, the second part of the objective might be more tedious. Indeed, to perform an accurate inference, we need to rely on a complicated non-linear function, which cannot be assumed to be Gaussian. To address this issue, we introduce the idea of *regression flows*. We consider that the transform $f_{\boldsymbol{\psi}(\mathbf{z})}$ is a normalizing flow (see Section 2.1) and provides two different ways of optimizing this approximation.

### 3.2.1. Posterior Parameterization

First, we follow a reasoning akin to the original formulation of normalizing flows by parameterizing the posterior $q_\phi(f_{\boldsymbol{\psi}}|\mathbf{z}, \mathbf{v})$ with a flow to obtain $q_k(\mathbf{v}_k)$. Hence, by developing the KL expression, we obtain

$$\mathcal{D}_{KL} \left[ q_\phi(f_{\boldsymbol{\psi}}|\mathbf{z}, \mathbf{v}) \| p(f_{\boldsymbol{\psi}}|\mathbf{z}) \right] = \mathbb{E}_{q_0} \left[ \log q_0(\mathbf{v}_0) \right] - \mathbb{E}_{q_0} \left[ \log p(\mathbf{v}_k) \right] - \mathbb{E}_{q_0} \left[ \sum_{i=1}^{k} \log \left| \det \frac{\partial f_i}{\partial \mathbf{v}_{i-1}} \right| \right]$$

Hence, we can now safely rely on Gaussian priors for $q_0(\mathbf{v}_0)$ and $p(\mathbf{v}_k)$. This formulation allows to consider $\mathbf{v}$ as a transformed version of $\mathbf{z}$, while being easily invertible as $\mathbf{z} = f_{[k,1]}^{-1}(\mathbf{v})$. We denote this version as $Flow_{post}$.

### 3.2.2. Conditional Amortization

Here, we consider that the parameters $\boldsymbol{\psi}$ of the flow are random variables that are optimized by decomposing the posterior KL objective as

$$\mathcal{D}_{KL} \left[ q_\phi(f_{\boldsymbol{\psi}}|\mathbf{z}, \mathbf{v}) \| p(f_{\boldsymbol{\psi}}|\mathbf{z}) \right] = \mathcal{D}_{KL} \left[ q_\phi(\boldsymbol{\psi}|\mathbf{z}) \| p(\boldsymbol{\psi}|\mathbf{z}) \right] + \mathbb{E}_{q_0(\mathbf{v}_0)} \left[ \sum_{i=1}^{k} \log \left| \det \frac{\partial f_i}{\partial \mathbf{v}_{i-1}} \right| \right]$$

As we rely on Gaussian priors for the parameters, this additional KL term can be computed easily. In this version, denoted $Flow_{cond}$, parameters of the flow are sampled from their distributions before computing the resulting transform.

*3.3. Disentangling Flows for Semantic Dimensions*

Based on the previous formulation, we can reintroduce the *semantic tags* in the model by expanding latent factors $\mathbf{z}$ with a categorical variable $\mathbf{t}$. Hence, we define the generative process $p_\theta(\mathbf{x}|\mathbf{t}, \mathbf{z})$ where $p(\mathbf{t}) = Cat(\mathbf{t}|\boldsymbol{\pi})$ and $p(\pi)$ is the prior distribution of the tags. We define the inference model as $q_\phi(\mathbf{z}, \mathbf{t}|\mathbf{x})$ and assume that it factorizes as $q_\phi(\mathbf{z}, \mathbf{t}|\mathbf{x}) = q_\phi(\mathbf{z}|\mathbf{x})q_\phi(\mathbf{t}|\mathbf{x})$. In order to handle the fact that tags are not always observed, we define a model similar to [18]. When $\mathbf{t}$ is unknown, it is considered as a latent variable over which we can perform posterior inference

$$\mathcal{L}_u = -\mathbb{E}\left[\log p_\theta(\mathbf{x}|\mathbf{t}, \mathbf{z}) + \log p_\theta(\mathbf{t}) + \log p_\theta(\mathbf{z})\right] - \mathbb{E}\left[\log q_\phi(\mathbf{t}, \mathbf{z}|\mathbf{x})\right]$$

When tags $\mathbf{t}$ are known, we take a rather unusual approach through the idea of *disentangling flows*. As we seek to obtain a latent dimension with continuous semantic control, we define a tag pair as a set of negative $\mathbf{t}_-$ and positive $\mathbf{t}_+$ samples. We define two *target* distributions $p(z_{\mathbf{t}_-}) \sim \mathcal{N}(-\mu_*, \sigma_-)$ and $p(z_{\mathbf{t}_+}) \sim \mathcal{N}(+\mu_*, \sigma_+)$ that model samples of a semantic pair as opposite sides of a latent dimension. Hence, we turn the treatment of tags into a *density estimation* problem, where we aim to match tagged samples $\mathbf{t}_*$ densities to given explicit target densities by minimizing $\mathcal{D}_{KL}\left[q_\phi(z_{\mathbf{t}_*}|\mathbf{x})\|p(z_{\mathbf{t}_*})\right]$. To solve this, we consider that $q_\phi(z_{\mathbf{t}_*}|\mathbf{x})$ is parameterized by a normalizing flow $f_k$ applied to the latent $\mathbf{z}$, leading to our final objective

$$\mathcal{L}_o = \mathcal{D}_{KL}\left[q_\phi(z_{\mathbf{t}_*})\|p(z_{\mathbf{t}_*})\right] = \mathbb{E}\left[\log p(\mathbf{z}) - \sum_{i=1}^{k}\log\left|\det\frac{\partial f_i}{\partial \mathbf{z}_{i-1}}\right| - \log p(z_{\mathbf{t}_*})\right] \qquad (13)$$

This formulation enforces a form of *supervised* disentanglement, where some of the latent dimensions $\mathbf{z}$ are transformed to provide controls with explicit semantic target properties. The final bound is defined as the sum of both objectives $\mathcal{L} = \mathcal{L}_o + \mathcal{L}_u$ and the complete model is obtained by integrating *regression* and *disentangling* flows together.

## 4. Experiments

*4.1. Dataset*

4.1.1. Synthesizer Sounds Dataset

We constructed a dataset of synthesizer sounds and corresponding parameters, by using an off-the-shelf commercial VST synthesizer *Diva* developed by U-He (https://u-he.com/products/diva/). It should be noted that our model can hypothetically work for any synthesizer, as long as we can produce couples of (audio, parameters) as input. We selected *Diva* as (i) almost all its parameters can be MIDI-controlled, (ii) large banks of presets are readily available, and (iii) presets include well-organized semantic tags pairs. The factory presets for Diva and additional presets from the internet were collected, leading to a total of roughly 11k files. We manually established the correspondence between synth and MIDI parameters as well as the parameters values range and their distributions. We only kept continuous parameters and normalize each of these parameters so that their values lie in the range $[0, 1]$. All other parameters are set to their fixed *default* value. Finally, we performed parameter selection by computing the PCA of the parameters value for the whole presets dataset. We sorted the contribution of each parameter to the principal components that explain more than 80% of the variance and performed manual screening to select increasing sets of the most used 16, 32 and 64 parameters. We use *RenderMan* (https://github.com/fedden/RenderMan) to batch-generate all the audio files by playing a C4 note for 3 s. and recording for 4 s. to capture the release of the note. The files are saved in 22,050 Hz and 16-bit floating point format.

### 4.1.2. Audio Processing

For each sample, we compute a 128 bins Mel-spectrogram with a FFT of size 2048 ms. with a hop of 1024 ms. and frequency range of $[30, 11,000]$ Hz. We only keep the magnitude of the spectrogram and perform a log-amplitude transform. The dataset is randomly split between a training (80%), validation (10%), and test (10%) set before each training. We repeat the training $k = 5$ times to perform $k$-fold cross-validation. Finally, we perform a corpus-wide zero-mean unit-variance normalization over the whole spectrogram based on the train set.

### 4.1.3. Metadata

Diva presets often contain useful metadata tags called *characteristics* that define high-level semantic properties of the audio output. Interestingly, these are well organized and defined as opposite pairs with clear concepts such as [*Bright*, *Dark*] or [*Soft*, *Aggressive*]. We retained a set of 10 such pairs and add the *Unknown* category for each pair when no tag of the pair is present (as presets may have multiple characteristics). Therefore, the final dataset is composed of triplets containing (synthesized audio output, parameters vector, semantic tags metadata).

### *4.2. Models*

### 4.2.1. Baseline Models

In order to evaluate our proposal, we implemented several feed-forward deep models that take the complete spectrogram $x_i$ of a sample as input and try to infer the corresponding parameters $v_i$. All these models are trained with a Mean-Squared Error (*MSE*) loss between the output of the model and the parameters vector. First, we implement a 5-layers *MLP* with 2048 hidden units per layer, Exponential Linear Unit (ELU) activation, batch normalization and dropout with $p = 0.3$. This model is applied on a flattened version of the input and the final layer is a sigmoid activation. We implement a convolutional model composed of 5 layers with 128 channels of strided dilated 2-D convolutions with kernel size 7, stride 2, and an exponential dilation factor of $2^l$ (starting at $l = 0$) with batch normalization and ELU activation. The convolutions are followed by a 3-layers MLP of 2048 hidden units with the same properties as the previous model. Finally, we implemented a *Residual Network* (denoted *ResCNN*), with parameters settings identical to *CNN*, while the residual paths are defined as simple $1x1$ convolution that maps to the same size.

### 4.2.2. Our Proposal

We implemented various *AE architectures, which are defined through two training losses. First, the traditional AE training is performed by using a *MSE* reconstruction loss on the spectrograms. We use the previously described *CNN* setup for both encoders and decoders. However, we halve their number of parameters (by dividing the number of units and channels) to perform a fair comparison by obtaining roughly the same capacity as the baselines. All AEs map to latent spaces of dimensionality equal to the number of synthesis parameters. For all these architectures, a second network is used to try to infer the parameters $v_i$ based on the latent code $z_i$ obtained by encoding a specific spectrogram $x_i$. For this part, we train all simple AE models with a 2-layers MLP of 1024 units to predict the parameters based on the latent space, with a *MSE* loss. First, we implement a simple deterministic *AE* without regularization. We implement the *VAE* by adding a KL regularization to the latent space and the *WAE* by replacing the KL by the MMD. Finally, we implement $VAE_{flow}$ by adding a normalizing flow of 16 successive IAF transforms to the *VAE* posterior. We perform *warmup* [11] by linearly increasing the latent regularization $\beta$ from 0 to 1 for 100 epochs. Then, we use *regression flows* ($Flow_{reg}$) by adding them to $VAE_{flow}$, with an IAF of length 16 without tags. In both cases, we introduce the regression objective only after 100 epochs and also apply warmup. Finally, we add the *disentangling flows* ($Flow_{dis}$) by adding our objective defined in Section 3.3.

### 4.2.3. Optimization Aspects

We train all models for 500 epochs with the ADAM optimizer, an initial learning rate of 0.0002, Xavier initialization of the weights and a scheduler that halves the learning rate if the validation loss stalls for 20 epochs. With this setup, the complete model ($VAE_{flow}$ with regression) only needs 5 h to complete training on a NVIDIA Titan Xp GPU.

## 5. Results

### 5.1. Parameters Inference

First, we compare the accuracy of all models on the *parameters inference* task by computing the magnitude-normalized *Mean Square Error* ($MSE_n$) between predicted and original parameters values. We average these results across folds and report variance. We also evaluate the distance between the audio synthesized from the inferred parameters and the original audio with the *Spectral Convergence* (SC) distance (magnitude-normalized Frobenius norm) and *MSE* (it should be noted that these measures only provide a global evaluation of spectrogram similarity, and that perceptual aspects of the results should be evaluated in human listening experiments that are left for future work). We provide evaluation results for 16, 32, and 64 parameters on the test set in Table 1.

**Table 1.** Comparison between baselines, *AEs, and our *flows* on the test set with 16, 32, and 64 parameters. We report across-folds mean and variance for parameters (Mean-Squared Error [$MSE_n$]) and audio (Spectral Convergence [$SC$] and $MSE_n$) errors. The best results are indicated in bold.

| | Test Set—16 Parameters | | | Test Set—32 Parameters | | | Test Set—64 Parameters | | |
|---|---|---|---|---|---|---|---|---|---|
| | Params | Audio | | Params | Audio | | Params | Audio | |
| | $MSE_n$ | $SC$ | $MSE_n$ | $MSE_n$ | $SC$ | $MSE_n$ | $MSE_n$ | $SC$ | $MSE_n$ |
| *MLP* | $0.236 \pm 0.44$ | $6.226 \pm 0.13$ | $9.548 \pm 3.1$ | $0.218 \pm 0.46$ | $13.51 \pm 3.1$ | $36.48 \pm 11.9$ | $0.185 \pm 0.41$ | $39.59 \pm 6.7$ | $49.58 \pm 2.7$ |
| *CNN* | $0.171 \pm 0.45$ | $1.372 \pm 0.29$ | $6.329 \pm 1.9$ | $0.159 \pm 0.46$ | $19.18 \pm 4.7$ | $33.40 \pm 9.4$ | $0.202 \pm 0.37$ | $52.48 \pm 7.2$ | $76.13 \pm 8.9$ |
| *ResNet* | $0.191 \pm 0.43$ | $1.004 \pm 0.35$ | $6.422 \pm 1.9$ | $0.196 \pm 0.49$ | $10.37 \pm 1.8$ | $31.13 \pm 9.8$ | $0.248 \pm 0.43$ | $29.18 \pm 3.8$ | $78.15 \pm 9.8$ |
| *AE* | $0.181 \pm 0.40$ | $0.893 \pm 0.13$ | $5.557 \pm 1.7$ | $0.169 \pm 0.40$ | $5.566 \pm 1.2$ | $17.71 \pm 6.9$ | $0.189 \pm 0.37$ | $8.123 \pm 2.4$ | $34.07 \pm 2.4$ |
| *VAE* | $0.182 \pm 0.32$ | $0.810 \pm 0.03$ | $4.901 \pm 1.4$ | $0.153 \pm 0.34$ | $5.519 \pm 1.4$ | $16.85 \pm 6.1$ | $0.171 \pm 0.37$ | $5.152 \pm 1.1$ | $33.10 \pm 2.4$ |
| *WAE* | $\mathbf{0.159 \pm 0.37}$ | $0.787 \pm 0.05$ | $4.979 \pm 1.5$ | $\mathbf{0.147 \pm 0.33}$ | $3.967 \pm 0.88$ | $16.64 \pm 6.2$ | $\mathbf{0.167 \pm 0.36}$ | $8.960 \pm 1.8$ | $\mathbf{32.59 \pm 2.1}$ |
| $VAE_{flow}$ | $0.199 \pm 0.32$ | $0.838 \pm 0.02$ | $4.975 \pm 1.4$ | $0.164 \pm 0.34$ | $1.418 \pm 0.23$ | $17.74 \pm 6.8$ | $0.174 \pm 0.36$ | $6.721 \pm 1.4$ | $33.81 \pm 2.3$ |
| $Flow_{reg}$ | $0.197 \pm 0.31$ | $\mathbf{0.752 \pm 0.05}$ | $\mathbf{4.409 \pm 1.6}$ | $0.193 \pm 0.32$ | $\mathbf{0.911 \pm 1.4}$ | $\mathbf{16.61 \pm 7.4}$ | $0.178 \pm 0.37$ | $\mathbf{4.794 \pm 1.8}$ | $34.49 \pm 2.2$ |
| $Flow_{dis.}$ | $0.199 \pm 0.31$ | $0.831 \pm 0.04$ | $5.103 \pm 2.1$ | $0.197 \pm 0.42$ | $1.481 \pm 1.8$ | $17.12 \pm 7.9$ | $0.182 \pm 0.38$ | $8.122 \pm 1.8$ | $34.97 \pm 2.3$ |

In low parameters settings, baseline models seem to perform an accurate approximation of parameters, with the *CNN* providing the best inference. Based on this criterion solely, our formulation would appear to provide only a marginal improvement, with *VAEs* even outperformed by baseline models and best results obtained by the *WAE*. However, analysis of the corresponding audio accuracy tells an entirely different story. Indeed, AEs approaches strongly outperform baseline models in audio accuracy, with the best results obtained by our proposed $Flow_{reg}$ (1-way ANOVA $F = 2.81$, $p < 0.003$). These results show that, even though AE models do not provide an exact parameters approximation, they are able to account for the importance of these different parameters on the synthesized audio. This supports our original hypothesis that learning the latent space of synthesizer audio capabilities is a crucial component to understand its behavior. Finally, it appears that adding disentangling flows ($Flow_{dis}$) slightly impairs the audio accuracy. However, the model still outperform most approaches, while providing the huge benefit of explicit semantic macro-controls.

### 5.2. Increasing Parameters Complexity

We evaluate the robustness of different models by increasing the number of parameters from 16 to 32 and finally 64 (Table 1). As we can see, the accuracy of baseline models is highly degraded, notably on audio reconstruction. Interestingly, the gap between parameter and audio accuracies is strongly increased. This seems logical as the relative importance of parameters in larger sets provoke stronger

impacts on the resulting audio. Also, it should be noted that *VAE∗* models now outperform baselines even on parameters accuracy. Although our proposal also suffers from larger sets of parameters, it appears as the most resilient and can still cope with this higher complexity. While the gap between AE variants is more pronounced, the *flows* strongly outperform all methods ($F = 8.13$, $p < 0.001$).

### 5.3. Reconstructions and Latent Space

We provide an in-depth analysis of the relations between inferred parameters and corresponding synthesized audio to support our previous claims. First, we selected two samples from the test set and compare the inferred parameters and synthesized audio in Figure 3.

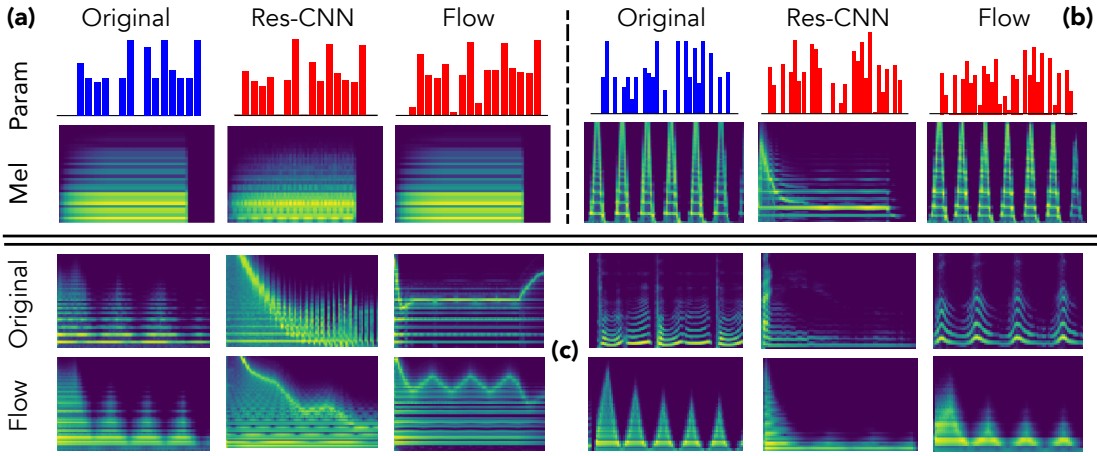

**Figure 3.** *Reconstruction analysis*. Comparing parameters inference and resulting audio on the test set with 16 (**a**) or 32 (**b**) parameters, and on the *out-of-domain* (**c**) sets composed either of sounds from other synthesizers (**left**) or vocal imitations (**right**).

As we can see, although the *CNN* provides a close inference of the parameters, the synthesized approximation completely misses important structural aspects, even in simpler instances as the simple harmonic structure in the first example (a). This confirms our hypothesis that direct inference models are unable to assess the *relative impact* of parameters on the audio. Indeed, the errors in all parameters are considered equivalently, even though the same error magnitude on two different parameters can lead to dramatic differences in the synthesized audio. Oppositely, even though the parameters inferred by our proposal are quite far from the original preset, the corresponding audio is largely more similar. This indicates that the latent space provides knowledge on the *audio-based neighborhoods* of the synthesizer. Therefore, this allows to understand the impact of different parameters in a given region of the latent audio space.

To confirm this hypothesis, we encode two random distant examples from the test set in the latent audio space and perform random sampling around these points to evaluate how local neighborhoods are organized. We also analyze the latent interpolation between those examples. The results are displayed in Figure 4. As we can see, our hypothesis seems to be confirmed by the fact that neighborhoods are highly similar in terms of audio but have a larger variance in terms of parameters. Interestingly, this leads to complex but smooth non-linear dynamics in the parameters interpolation.

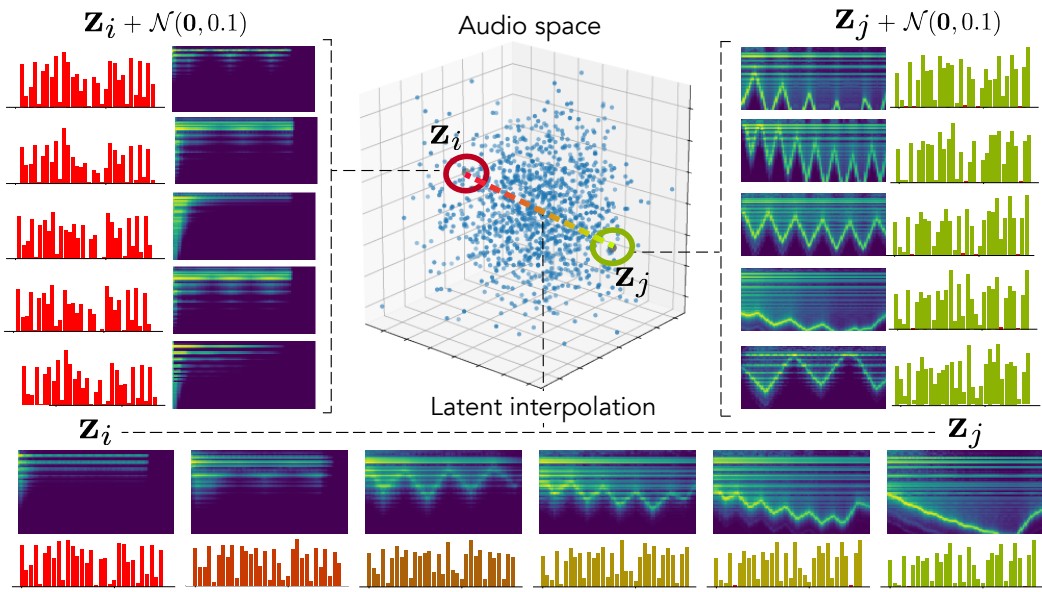

**Figure 4.** *Latent neighborhoods.* We select two examples from the test set that map to distant locations in the latent space **z** and perform random sampling in their local neighborhood to observe the parameters and audio. We also display the latent interpolation between those points.

### 5.4. Out-Of-Domain Generalization

We evaluate *out-of-domain generalization* by applying parameters inference and re-synthesis on two sets of audio samples either produced by other synthesizers, or with vocal imitations. We rely on the same evaluation method as previously described and provide results for the audio similarity in Table 2 (Right). Here, the overall distribution of scores remains consistent with previous observations. However, it seems that the average error is quite high, indicating a potentially distant reconstruction of some examples. This might be explained by the limited number of parameters used for training our models. Therefore, they cannot account for complex sounds with various types of modulations. Interestingly, while the addition of more parameters to perform the optimization allows to reduce the global approximation error in AE models, it seems to worsen the feed-forward estimation. This seems to further confirm our original hypothesis that feed-forward approaches are not able to handle advanced interactions in the parameters.

**Table 2.** Comparison between baselines, *AEs, and our *flows* on the out-of-domain parameters inference task. We report across-folds mean and variance for parameters (MSE) and audio (SC and MSE) errors.

|  | Out-of-Domain (32 p.) | | Out-of-Domain (64 p.) | |
|---|---|---|---|---|
|  | **SC** | **MSE** | **SC** | **MSE** |
| *MLP* | $2.348 \pm 2.1$ | $37.99 \pm 7.8$ | $4.534 \pm 5.1$ | $40.42 \pm 3.7$ |
| *CNN* | $2.311 \pm 2.2$ | $29.22 \pm 8.2$ | $6.329 \pm 1.9$ | $36.93 \pm 2.3$ |
| *ResNet* | $2.322 \pm 1.6$ | $31.07 \pm 9.5$ | $4.645 \pm 3.1$ | $27.46 \pm 2.3$ |
| *AE* | $1.225 \pm 2.2$ | $27.37 \pm 7.2$ | $2.557 \pm 1.7$ | $27.16 \pm 1.4$ |
| *VAE* | $1.237 \pm 1.3$ | $27.06 \pm 7.1$ | $1.141 \pm 1.2$ | $27.15 \pm 1.3$ |
| *WAE* | $\mathbf{1.194 \pm 1.5}$ | $26.10 \pm 6.4$ | $\mathbf{0.999 \pm 0.9}$ | $25.13 \pm 1.3$ |
| *VAE$_{flow}$* | $1.193 \pm 1.8$ | $27.03 \pm 6.4$ | $1.022 \pm 1.7$ | $26.49 \pm 1.3$ |
| *Flow$_{reg}$* | $1.201 \pm 1.2$ | $\mathbf{26.07 \pm 7.7}$ | $1.132 \pm 1.6$ | $\mathbf{24.74 \pm 1.3}$ |
| *Flow$_{dis.}$* | $1.209 \pm 1.4$ | $26.77 \pm 7.3$ | $1.532 \pm 1.8$ | $27.89 \pm 1.7$ |

In order to better understand the results and limits of our proposal, we display in Figure 3 the resynthesis of random examples taken from the synthesizer (left) and vocal imitations (right) datasets. As we can see, in all cases, our proposal accurately reproduces the temporal spectral shape of target sounds, even if the timbre is somewhat distant. Upon closer listening, it seems that the models fail to reproduce the local timbre of voices but performs quite well with sounds from other synthesizers. However, the evolution of the spectral shape is still reproduced. Interestingly, this provides a form of *vocal sketching control* where the user inputs vocal imitations of the sound that he is looking for. This allows to quickly produce an approximation of the intended sound and, then, exploring the audio neighborhood of the sketch for intuitive refinement.

### 5.5. Macro-Parameters Learning

Our formulation is the first to provide a continuous mapping between the audio **z** and parameter **v** spaces of a synthesizer. As latent VAE dimensions has been shown to disentangle major data variations, we hypothesized that we could directly use **z** as *macro-parameters* defining the principal dimensions of audio variations in a given synthesizer. Hence, we introduce the new task of *macro-parameters learning* by mapping latent audio dimensions to parameters through $p(\mathbf{v}|\mathbf{z})$, which provides simplified control of the major audio variations for a given synthesizer. This is depicted in Figure 5.

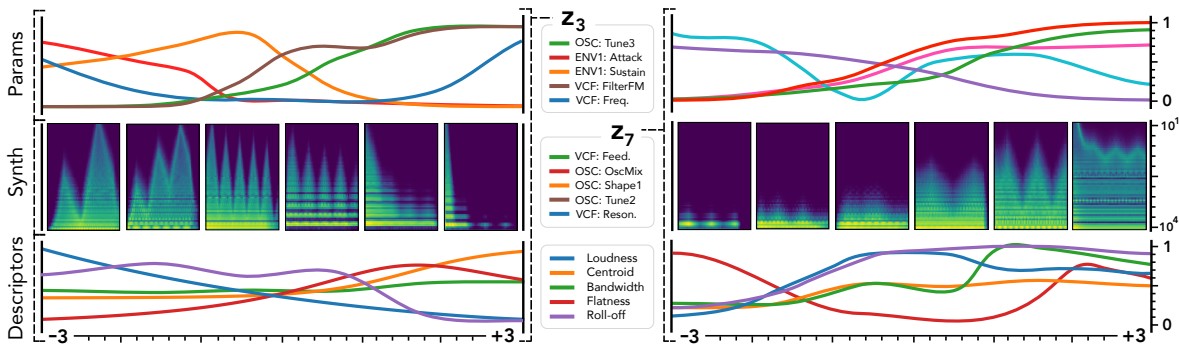

**Figure 5.** *Macro-parameters learning.* We show two of the learned latent dimensions **z** and compute the mapping $p(\mathbf{v}|\mathbf{z})$ when traversing these dimensions, while keeping all other fixed at **0** to see how **z** define smooth macro-parameters. We plot the evolution of the 5 parameters with highest variance (**top**), the corresponding synthesis (**middle**), and audio descriptors (**bottom**). (**Left**) $\mathbf{z}_3$ seems to relate to a *percussivity* parameter. (**Right**) $\mathbf{z}_7$ defines a form of *harmonic densification* parameter.

We show the two most informative latent dimensions **z** based on their variance. We study the traversal of these dimensions by keeping all other fixed at **0** to assess how **z** defines smooth macro-parameters through the mapping $p(\mathbf{v}|\mathbf{z})$. We report the evolution of the 5 parameters with highest variance (top), the corresponding synthesis (middle) and audio descriptors (bottom).

First, we can see that latent dimension corresponds to very smooth evolutions in terms of synthesized audio and descriptors. This is coherent with previous studies on the disentangling abilities of VAEs [6]. However, a very interesting property appear when we map to the parameter space. Although the parameters evolution is still smooth, it exhibits more non-linear relationships between different parameters. This correlates with the intuition that there are lots of complex interplays in parameters of a synthesizer. Our formulation allows to alleviate this complexity by automatically providing *macro-parameters* that are the most relevant to the audio variations of a given synthesizer. Here, we can see that the $\mathbf{z}_3$ latent dimension (left) seems to provide a *percussivity* parameter, where low values produce a very slow attack, while moving along this dimension, the attack becomes sharper and the amount of noise increases. Similarly, $\mathbf{z}_7$ seems to define an *harmonic densification* parameter, starting from a single peak frequency and increasingly adding harmonics and noise. Although the unsupervised macro-parameters provide some clear effects on the synthesis, it appears that they do

not act on a single aspect of the timbre. This seems to indicate that the macro-parameters still relate to some entangled properties of the audio. Furthermore, as these dimensions are unsupervised, we still need to define their effects through direct exploration. Additional macro-parameters are discussed on the supporting webpage of this paper.

### 5.6. Semantic Parameter Discovery

Our proposed *disentangling flows* can steer the organization of selected latent dimensions so that they provide a separation of given tags. As this audio space is mapped to parameters through $p(\mathbf{v}|\mathbf{z})$, this turns the selected dimensions into *macro-parameters* with a defined semantic meaning. To evaluate this, we analyze the behavior of corresponding latent dimensions, as depicted in Figure 6.

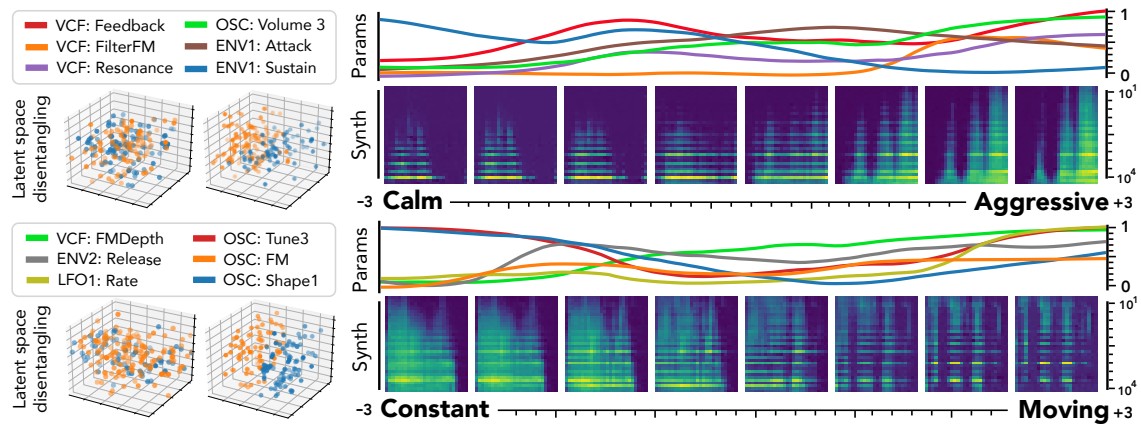

**Figure 6.** *Semantic macro-parameters*. Two latent dimensions $\mathbf{z}$ learned through *disentangling flows* for different pairs. We show the effect on the latent space (**left**) and parameters mapping $p(\mathbf{v}|\mathbf{z})$ when traversing these dimensions, that define smooth macro-parameters. We plot the evolution of 6 parameters with highest variance and the resulting synthesized audio (**right**).

First, we can see the effect of disentangling flows on the latent space (left), which provide a separation of semantic pairs. We study the traversal of semantic dimensions while keeping all other fixed at **0** and infer parameters through $p(\mathbf{v}|\mathbf{z})$. We display the 6 parameters with highest variance and the resulting synthesized audio. As previously observed for unsupervised dimensions, the semantic latent dimensions also seem to provide a very smooth evolution in terms of both parameters and synthesized audio. Regarding the precise effect of different semantic dimensions, it appears that the ['*Constant*', '*Moving*'] pair provides a very intuitive result. Indeed, the synthesized sounds are mostly stationary in extreme negative values, but gradually incorporate clearly marked temporal modulations. Hence, our proposal appears successful to uncover *semantic macro-parameters* for a given synthesizer. However, the corresponding parameters are quite harder to interpret. The ['*Calm*', '*Aggressive*'] dimension also provides an intuitive control starting from a sparse sound and increasingly adding modulation, resonance and noise. However, we note that the notion of '*Aggressive*' is highly subjective and requires finer analyses to be conclusive.

### 5.7. Creative Applications

Our proposal allows to perform a direct exploration of presets based on audio similarity. Indeed, as the flow is *invertible*, we can map parameters to the audio space for exploration, and then back to parameters to obtain a new preset. Furthermore, this can be combined with *vocal sketch control* where the user inputs vocal imitations of the sound that he is looking for. In order to allow creative experiments, we implemented all the models and interactions detailed in this paper in an experimental Max4Live interface that is displayed in Figure 7. We embedded our models inside *MaxMSP* by using an *OSC* communication server with the *Python* implementation. We further integrate it into *Ableton Live*

by using the *Max4Live* interface. This interface wraps the Diva VST and allows to provide control based on all of the proposed models. Hence, this interface allows to input a wave file or direct vocal recording to perform *parameter inference*. The model can provide the VST parameters for the approximation in less than 30 ms on a CPU. The interface also provides a representations of the projected latent audio space, onto which is plotted the preset library. This allows to perform audio-based preset exploration, but also to draw paths between different presets or simply across the audio space. By freely exploring the dimensions, the user can also experiment the *unsupervised macro-control* and also explore *supervised semantic dimensions*. Finally, we implemented an interaction with the *Leap Motion* controller, which allows to directly control the synthesized sound with one's hand.

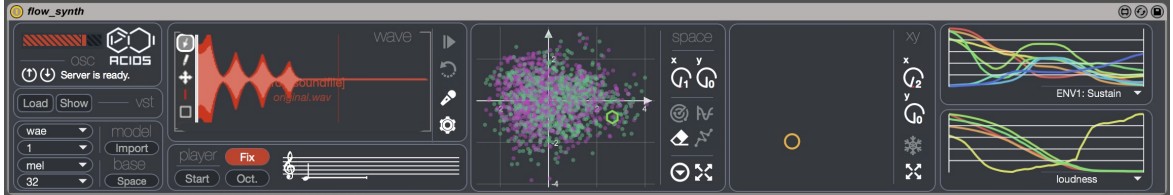

**Figure 7.** *FlowSynth* interface for audio synthesizer control in Ableton Live. The interface wraps a given VST, and allows to perform direct parameters inference, audio-based preset exploration and relying on both semantic and unsupervised macro-controls learned by our model.

## 6. Conclusions

In this paper, we discussed several novel ideas based on our recent novel formulation of the problem of synthesizer control as matching the two latent spaces defined as the *audio perception space* and the *synthesizer parameter space*. To solve this new formulation, we relied on VAEs and Normalizing Flows to organize and map the auditory and parameter spaces of a given synthesizer. We introduced the *disentangling* flows, which allow to obtain an invertible mapping between two separate latent spaces, while steering the organization of some latent dimensions to match target variation factors by splitting the objective as partial density evaluation.

We showed that our approach outperforms all previous proposals on the seminal problem of *parameters inference*, and that it is able to provide an interesting approximation to any type of sound in almost real-time, even on a CPU. We showed that for sounds that are not produced by synthesizers, our model is able to match the evolution of the spectral shape quite well, even though the local timbre is not well approximated. We further showed that our formulation also naturally introduces various original and first-of-kind tasks of *macro-control learning*, *audio-based preset exploration*, and *semantic parameters discovery*. Hence, our proposal is the first to be able to simultaneously address most synthesizer control issues at once, while providing higher-level understanding and controls. In order to allow for usable and creative exploration of our proposed methods, we implemented a Max4Live interface that is available freely along with the source code of all approaches on the supporting webpage of this paper.

Altogether, we hope that this work will provide new means of exploring audio synthesis, sparking the development of new leaps in musical creativity.

**Author Contributions:** Conceptualization, P.E. and A.B.; Data curation, N.M. and R.D.; Formal analysis, P.E. and A.C.-R.-S.; Funding acquisition, P.E.; Investigation, P.E., N.M., R.D. and A.C.-R.-S.; Methodology, P.E., N.M., A.B. and A.C.-R.-S.; Project administration, P.E.; Resources, A.B.; Software, P.E., N.M. and R.D.; Validation, A.C.-R.-S.; Visualization, N.M.; Writing—original draft, P.E. All authors have read and agreed to the published version of the manuscript.

**Funding:** This work was supported by the MAKIMOno project funded by the French ANR and Canadian NSERC (ANR:17-CE38-0015-01 and NSERC:STPG 507004-17) and also the ACTOR Partnership funded by the Canadian SSHRC (SSHRC:895-2018-1023). This work was also supported by an NVIDIA GPU Grant and GPU Center grant.

**Conflicts of Interest:** The authors declare no conflict of interest.

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
