# Peer review of "Flow Synthesizer: Universal Audio Synthesizer Control with Normalizing Flows†"

_applsci, doi:10.3390/app10010302_

Round 1

Reviewer 1 Report

This paper presents a very interesting work that can be of interest of many in the field. However, it is heavily based on a previous work by the authors titled "UNIVERSAL AUDIO SYNTHESIZER CONTROL WITH NORMALIZING FLOWS", that has been presented at the 22nd International Conference on Digital Audio Effects (DAFx-19), Birmingham, UK, September 2–6, 2019. Even though additional details are shared in the version submitted to this journal, there are a few things that the authors should consider reviewing, namely the abstracts and introductions of these two papers, the one presented at the conference and this one, are exactly the same, word by work. This is not a good practice, and the authors should consider revising this. Moreover, the authors should also cite the above mentioned conference paper and explain how this one relates it. What are the additional information or details that are presented in this paper that justifies its publication in this journal?

I would not recommend the publication of this paper if those issues are not properly addressed.

Author Response

We entirely agree with the reviewers comment. As we received a specific mail invitation from the DaFX Program Committee to submit an extended version of our DaFX paper to this special issue, we inferred (apparently wrongfully) that this issue was entirely dedicated to extended versions of the DaFX papers, and that the "extended" mentions would be handled by the editor rather than the authors of the different papers.

After checking different mails and instructions pages, we could not find any instruction on how this should be properly handled. So, we performed the following modifications based on the reviewer wise comments, and hope that the editor will confirm and/or correct our modified version to fit the requirements of this special issue.

1 - We modified the abstract to reflect the extended part of the manuscript. We added a footnote referenced right away in the abstract to state that this paper was an extended version of the DaFX one. We also rephrased large parts of the abstract to underline the differences between the conference paper and the novel contributions.

2 - We reworked the introduction by adding the reference to the DaFX paper at the onset of the model discussion. We then clearly underline the parts that were introduced in the DaFX paper, and those that are novel and detailed in the extended version.

We hope that the editor can confirm that this presentation will fit the editorial line of the journal. Also, we were not sure whether we should keep the exact same paper title, to simplify the connection between both versions of the paper (we added « Flow Synthesizer: » at the beginning of the original DaFX title, but would be glad to follow the editor advices on the best practice for this aspect).

Reviewer 2 Report

This paper is about inferring synthesizer parameters from synthesized audio with deep learning methods. This is an interesting research problem since there is a complex interplay between a music synthesizer's control parameters and the sounds produced, especially for synthesis methods like subtractive synthesis or FM. While the authors also sketch several creative application scenarios, I think the current products with pre-defined and curated presets will still be the prevailing paradigm. However, it is still worthwhile to study this task.

The core idea of the paper is to learn a latent space of the spectral and temporal features that the synthesizer can produce in an auto-encoder fashion. At the same time, an invertible mapping of this abstract representation to the parameter space of the synthesizer is performed. Additional ideas are the inclusion of semantic tags in the training to foster supervised disentangelement.

To evaluate their proposed approaches, the authors compile a dataset made of presets of a VST synthesizer. This dataset comes with pairs of audio samples and parameter settings as well as semantic tags of the presets. The audio representation used in the experiments is a mel-scaled log-magnitude spectrogram, the parameters to infer are based on different PCA projections of the original parameters.

The authors compare quite a range of different deep learning architectures against for the task of parameter inference. They have fully connected and convolutional baseline models. The next level are different flavours of auto-encoder and variational auto-encoder architectures. Finally, they add normalizing flows with different objectives to the VAEs. Their results show, that the VAE with regression flows almost always outperforms the other methods, while there is a strong dependency on the parameter set to be inferred.

The remainder of the paper explores the learned latent spaces, the ability to cope with unseen data, macro-parameter learning and practical implementation into DAW software.

In the following, I will list some open questions and points for improvement of the paper.

1) Math notation

in eq (4), the parameters \theta appear and are never really explained in seq 2.2, the symbol denoting real numbers changes from \mathbb{R} to \mathcal{R} for no particular reason Why not spend more time on explaining normalizing flows, they are still a pretty new concept

2) Experiments

sec 3.2.1. is very vague in its description of the PCA and manual screening used to define the sets of 16, 32, and 64 control parameters. What did you actually do here? the frequency range of your mel-spectrogram should be given in Hz and not unit-less please for convenience, also provide the blocksize and hopsize in ms you mention k-fold cross validation and report standard devation of your metrics but you never state what k was set to Along which dimension to you normalize the corpus (frames, mel-bands, or complete spectrograms?). Do you normalize the target control parameter sets as well? Where the semantic tags somehow balanced? Since you are using spectral convergence as a quality measure, you should mention that it is not a perceptual score and you should consider to give it in dB

3) Writing and presentation

section 2.3 is very similar in content to the introduction and could be shortened or removed the writing is very clear, I only spotted one type in the 1st sentence of seq 3.1, it should be int_r_oduce instead of intoduce You should merge fig 1 and fig 2, since fig 1b is only a compacted version of fig 2

Author Response

We would like to thank the reviewer for the very thorough and detailed comments on our paper to improve its quality. We discuss each point separately in the following.

1) Math notation

in eq (4), the parameters \theta appear and are never really explained in seq 2.2, the symbol denoting real numbers changes from \mathbb{R} to \mathcal{R} for no particular reason  > We introduced a clear explanation of the likelihood and prior p_\theta and corrected the typos for the real numbers   Why not spend more time on explaining normalizing flows, they are still a pretty new concept > We tried to extend the explanation around normalizing flows and spend more time discussing their properties. 

2) Experiments

sec 3.2.1. is very vague in its description of the PCA and manual screening used to define the sets of 16, 32, and 64 control parameters. What did you actually do here?  > We updated this part to provide a more detailed explanation of the selection procedure, which was indeed unclear.   the frequency range of your mel-spectrogram should be given in Hz and not unit-less please for convenience, also provide the blocksize and hopsize in ms  > Updated in the new manuscript   you mention k-fold cross validation but you never state what k was set to  > Updated in the new manuscript   Along which dimension to you normalize the corpus (frames, mel-bands, or complete spectrograms?).  > Updated in the new manuscript   Do you normalize the target control parameter sets as well?  > We discussed parameters normalization in the first version of the paper, but we rephrased it to be clearer.   Where the semantic tags somehow balanced?  > We did not select presets based on the semantic tags as the original dataset was already quite balanced (and also quite sparse)   Since you are using spectral convergence as a quality measure, you should mention that it is not a perceptual score > We are not sure that we fully understood that comment, as we never implied that the SC measure was a perceptual score in the original paper. However, we added a footnote at the beginning of Section 5.1 to underline this aspect.  

3) Writing and presentation

section 2.3 is very similar in content to the introduction and could be shortened or removed > We agree with the reviewer on this redundancy. However, we thought that it was more adequate to reduce the redundancy in the introduction (as it was already too long) to provide a clearer explanation, while keeping an extended discussion of the previous works in 2.3   The writing is very clear, I only spotted one typo in the 1st sentence of seq 3.1, it should be int_r_oduce instead of intoduce  > We thank the reviewer for his very thorough review of the paper, and corrected the typo   You should merge fig 1 and fig 2, since fig 1b is only a compacted version of fig 2  > In this case, we disagree with the reviewer, as Figure 1 allows to compactly compare the feedforward approach to ours. Also, we believe that the simplified aspect of Fig 1b provides the reader with a clearer schematic understanding of how our model is constructed.

Round 2

Reviewer 1 Report

I am glad to notice that the authors have addressed the small issues I mentioned in my earlier review. I therefore recommend the acceptance of this paper in its present form. Final minor editing review might be required, but I leave that to the discretion of the editors.